# Application of Bacteriophages on Shiga Toxin-Producing *Escherichia coli* (STEC) Biofilm

**DOI:** 10.3390/antibiotics10111423

**Published:** 2021-11-20

**Authors:** Nicola Mangieri, Roberto Foschino, Claudia Picozzi

**Affiliations:** DeFENS, Department of Food, Environmental and Nutritional Sciences, Università degli Studi di Milano, Via G. Celoria 2, 20133 Milano, Italy; nicola.mangieri@unimi.it (N.M.); roberto.foschino@unimi.it (R.F.)

**Keywords:** biofilm, bacteriophages, Shiga toxin-producing *Escherichia coli* (STEC), biocontrol

## Abstract

Shiga toxin-producing *Escherichia coli* are pathogenic bacteria able to form biofilms both on abiotic surfaces and on food, thus increasing risks for food consumers. Moreover, biofilms are difficult to remove and more resistant to antimicrobial agents compared to planktonic cells. Bacteriophages, natural predators of bacteria, can be used as an alternative to prevent biofilm formation or to remove pre-formed biofilm. In this work, four STEC able to produce biofilm were selected among 31 different strains and tested against single bacteriophages and two-phage cocktails. Results showed that our phages were able to reduce biofilm formation by 43.46% both when used as single phage preparation and as a cocktail formulation. Since one of the two cocktails had a slightly better performance, it was used to remove pre-existing biofilms. In this case, the phages were unable to destroy the biofilms and reduce the number of bacterial cells. Our data confirm that preventing biofilm formation in a food plant is better than trying to remove a preformed biofilm and the continuous presence of bacteriophages in the process environment could reduce the number of bacteria able to form biofilms and therefore improve the food safety.

## 1. Introduction

Shiga toxin-producing *Escherichia coli* (STEC) are characterized by the presence of virulence factors that can cause severe disease in humans such as Hemorrhagic Colitis (HC) and Hemolytic Uremic Syndrome (HUS) [1]. This heterogeneous group of pathogenic bacteria has been responsible for large outbreaks worldwide causing more than 1 million illnesses and 100 deaths in 2010 [2]. STEC infection was the third zoonosis with 7775 confirmed cases in Europe in 2019, with an increasing trend from 2015 to 2019. Foodborne outbreaks are mainly linked to consumption of bovine meat and products thereof, milk and tap water, including well water. Of the isolates from food with the reported information on the serogroup 21.6% belonged to the ‘top-five’ serogroups (O157, O26, O103, O111 and O145) and more than half of all the remaining STEC belonged to the top-20 STEC serogroups reported in human infections to ECDC in 2015–2018 [3].

Moreover, other STEC serogroups have been associated with HUS and other infections in humans and, after the large outbreaks in Germany in 2011, a new strain presenting an unusual combination of virulence factors of STEC and Enteroaggregative *E. coli* (EAggEC) has been identified [4]. This finding, together with the analysis of the confirmed reported human infections in Europe suggested that there is no single or combination of virulence traits that can be associated exclusively with severe illness and that, therefore, all STEC strains can be considered human pathogens [5].

The ability of STEC strains to form biofilms on abiotic surfaces as well as on different types of foods can be considered another risk factor. Bacterial biofilms can be described as microbial sessile communities in which bacterial cells are embedded in a matrix of polymeric substances that increase their adhesion to food and food surfaces, becoming a source of contamination that is difficult to remove [6] and making bacteria more resistant to sanitation treatment [7]. Indeed, biofilms containing STEC or other pathogenic microorganisms can contribute to their persistence in food systems increasing the risk of foodborne outbreaks [8,9,10] and can also be a perfect ecosystem for the dissemination of *stx* genes and therefore for the development of new pathogenic isolates [11,12].

Over the years several approaches for biofilm control and removal have been applied, mainly based on physical and chemical processes [13]; however, all these strategies were often ineffective due to the physical barrier made of extracellular polymeric substances (EPS), but also because the cells within biofilms can be less metabolically active and therefore less affected by chemical substances such as antibiotics [14]. Bacteriophages, as natural enemies of bacteria, can potentially be a valid alternative to counteract biofilm formation or to facilitate their removal. Phages have high host specificity and evolving capacity, are auto-replicative, do not affect the intestinal microbiota and, most of all, are able to produce specific enzymes (depolymerases) that can degrade the EPS matrix [15,16,17]. However, this strategy can also have a few drawbacks mainly related to the variability in the composition of biofilms, both in terms of structure (multiple layers) and of species and strains [18], and the emergence of phage resistance [19,20]. Nevertheless, phages are able to co-evolve with their hosts overcoming mechanism of resistance and this can be improved by the use of phage cocktails widening the host range [21].

In this work, STEC strains were studied for their capability to produce biofilm. Among the best producers, four strains were chosen to be tested against different single bacteriophages and two different cocktails of three or six bacteriophages.

## 2. Results

### 2.1. Biofilm Formation Assay

In TSB medium, 15 out of 31 (48%) of investigated strains showed at least weak (W) biofilm formation at 30 °C, while, 13 out of 31 formed at least W biofilm at 37 °C, according to Naves et al. [22]. In M9 medium supplemented with 0.5% glucose a significantly fewer strains were able to form biofilm, namely 12.9% at 30 °C and 3.2% at 37 °C. 32% of investigated strains did not produce detectable biofilm under any of the conditions investigated (Table 1). F95 and C679-12 showed Strong (S) biofilm production, at least in one condition performed. 

### 2.2. Effects of Bacteriophages on Biofilm Prevention 

The capability of bacteriophages to prevent the formation of biofilms produced by STEC strains was investigated. The bacterial strains used in the experiments were selected from those that showed biofilm formation in crystal violet assay and in preliminary testing on biofilm production on a membrane. We used ED56, C679-12, ED226 and ED33 STEC strains and two non-pathogenic *E. coli* strains as the control (CNCTC6246 and ACTC25404). The biofilm prevention test was performed by comparing the attached cells of bacteria grown in presence of bacteriophage, used alone or in a cocktail, to the control. Observing the OD_600nm_ values, it is possible to note that the mean value of the control is 0.304, while the value obtained after bacteriophage treatment at different Multiplicities of Infection (MOI) is 0.173, thus highlighting a reduction in attached cells of 43.09%. Each single MOI showed a different level of reduction. The least effective was MOI 2 with a reduction of 32.24%, followed by MOI 10 and 1 with a reduction of and 45.07% and 45.40%, respectively. MOI 100 appeared the most effective with a reduction of 50.33% (Figure 1). 

Furthermore, the six bacteriophages tested showed an average reduction between 37.83% and 42.48%; the cocktails proved to be more effective with a reduction of 47.04% and 48.35% for cocktails composed of 3 and 6 phages, respectively. Therefore, the cocktails did not show a significative difference compared to single phages (Figure 2). 

As for the phages used alone, the most effective were DP17 and FM10 that showed a similar value (Table 2) and a 45% reduction in the biofilm formation. 

Each bacterial strain showed a particular behavior: ED33 (O139) was the most sensitive, while ED226 (0113) was the most resistant. In fact, with all the MOI values used, the reduction in biofilm production for ED33 was on average 59.14%, where the MOI 100 showed greater efficacy, with a reduction of up to 64.10% (Appendix A). In this case, there was no particular difference among the phages used.

A large decrease in biofilm production also occurred in the two non-pathogenic *E. coli* strains: CNCTC 6246 and ATCC 25404. The first showed a 68.04% reduction in biofilm production, while the second of 51.68%. These results showed that when there is a strong reduction in the biofilm there are no significant differences among the different MOI used. Conversely, a significant reduction in biofilm production for C679-12 strain was achieved only with MOI 100, leading to a decrease of 60.78%. Strain ED226 showed the lower reduction in biofilm formation, with an average MOI value of 24.24%. An interesting behavior was highlighted for the ED 56 strain, where the presence of the bacteriophage in the growth medium increased biofilm production by 44.15%.

### 2.3. Effects of Bacteriophages on Already Formed Biofilms

To remove the already formed biofilm, the six-phage cocktail, which was demonstrated to have a slightly better performance in biofilm prevention, was used against a biofilm formed in 24 h by four different STEC strains belonging to different serogroups (O26, O104, O113, O139) and one high biofilm producer *E. coli* strain. This assay, used for testing antimicrobial activity against static biofilm through the enumeration of cultivable cells, allows attribution of the result to cell death rather than detachment [25]. For the evaluation of the results, the values of CFU/cm^2^ of *E. coli* strains in membranes inoculated with phages were compared with the ones of the controls, which are membranes with the *E. coli* strain but without phages. 

In this case, the application of the phage cocktail on pathogenic *E. coli* biofilms did not reduce the number of biofilm-forming cells. The results showed no significant differences (*p* < 0.05) between the control and samples treated with the bacteriophage cocktail for all bacterial strains used as targets (Figure 3). 

### 2.4. Effects of Bacteriophages on Stx Production

Figure 4 shows the effect of exposure to Cocktail 2 on Stx production. Concerning strain ED56, the presence of Mitomycin C induced a 2.5-fold increase in toxin release compared to non-exposed culture while exposure to bacteriophages did not induce Stx production. On the other side, strain ED226 was not affected by Mitomycin C nor by phage cocktail showing a negative value of fold change.

## 3. Discussion

In the food industry, biofilm formation can pose a safety hazard, especially when it forms on surfaces that come in contact with food. One of the main risk is due to the detachment of cells from biofilm matrix and the resulting contamination of food [26]. Hence, the impact on human health and the economic losses have promoted the development of different approaches to control or remove the biofilms. In recent years, bacteriophages were applied as a tool to reduce biofilm formation in various pathogenic bacteria, such as *Pseudomonas aeruginosa* [27,28,29,30,31], *Klebsiella pneumoniae* [32], *E. coli* [33,34,35,36,37], and *Staphylococcus epidermidis* [38]. Thus, bacteriophages were tested in this study to treat biofilms produced by STEC strains. 

The bacteria used in the experiments were selected from those that showed the highest biofilm formation under the chosen experimental conditions. Our results confirm that formation of biofilm by STEC strains on polystyrene surfaces is heterogeneous and strongly dependent on strain rather than serotype [7].

Furthermore, treatments with both the cocktails did not show significative differences compared to the single phage when applied for the prevention of biofilm formation. This is probably due to the fact that use of the same site of viral attachment on the bacterial surface limits a simultaneous attack by multiple phages. However, we consider a cocktail the best option, since biofilms can often harbor different species and strains of bacteria meaning that a mixture of different viruses can be more efficient in controlling biofilms. Furthermore, having more phages that are active on a strain can prevent the emergence of a phage-resistant bacterium.

It seems rather strange that the presence of bacteriophages could increase the production of biofilm, as occurred in our experiments. However, different authors confirmed that phage predation can lead to increased bacterial biofilm levels. Various mechanisms have been proposed to elucidate this phenomenon, such as the strong selective pressure that virulent phages exert on their host community that can lead to resistant mutants [39], the phage-mediated cell lysis that trigger the release of crucial biofilm promoting factors [40], or a bacterial *quorum-sensing* system that may induce production of biofilms [41]. However, using multiple phages at the same time has been shown to reduce the frequency of resistance in the bacterial host, so the emergence of phage resistance was not the main phenomenon limiting the efficiency of phage processing in controlling biofilms. The study of Lacqua et al. [39] suggest that cell growth as a biofilm can produce a complementary or synergistic mechanism of resistance to bacteriophages with more specific processes such as the loss of phage receptors or the expression of DNA restriction enzymes, thus indicating a close evolutionary relationship between the ability to form a biofilm and the resistance to phages.

However, our data indicate that our bacteriophages can be useful in preventing the formation of biofilm since they were able to reduce biofilm formation by 43.09% on average with a slightly better performance of the cocktail containing six bacteriophages. 

This cocktail was then used to counteract pre-formed biofilms. The statistical analysis revealed no significant differences in any of the strains analyzed between the control and the treated biofilm and this fact can be influenced by different factors. A study on biofilm biocontrol of Jassim et al. [42] underlined the importance of the phage application method in terms of relevant concentration, duration of exposure, post-application storage temperature and type of samples treated. Kelly and colleagues [43] found that biofilm formed by *Staphylococcus aureus* was reduced significantly after 48 and 72 h, but not after 24 h of exposure. Therefore, contact time can be one of the causes of the absence of significative reduction. However, the application of our phages is intended for the food industry where it is unthinkable to stop a plant for 48 or even 72 h. 

A possible further explanation may be related to the fact that the bacteriophages used in this study were unable to access the biofilm and reach the bacteria within its matrix, inducing cell death. Access to the biofilm matrix in this case should be assisted by bacteriophage depolymerases, which are thought to be responsible for the degradation of EPS [18]. 

As already stated, when the contact between bacterial cells producing biofilm and bacteriophages started from the beginning of the experiment, it is possible to note a significative reduction, while the old biofilm is difficult to remove and in general less favorable to phage diffusion [18]. Thus, the nature of biofilms, the ability of *E. coli* to adhere to surfaces and the problems associated with their treatment and removal indicate that biofilm prevention is preferable to biofilm destruction and removal [44].

The semi-quantitative approach used to calculate free Stx in the supernatants exposed to phage cocktails showed that there is no increase in toxin release compared to the supernatant of non-exposed cultures. These results demonstrate that the use of phages does not undermine food safety.

In conclusion, the data reported in this paper demonstrated that bacteriophages can prevent the formation of biofilms in verotoxic *E. coli* strains. However, these phages were unable to reduce the population of the pre-formed biofilms. This result further underlines that, once formed, the biofilm is more difficult to eradicate and remove. Therefore, prevention could be the most suitable solution in the food industry and having bacteriophages that remain in the processing environment could lead to a continuous inactivation of those bacteria that can form biofilms.

## 4. Materials and Methods

### 4.1. Bacterial Cultures

The bacteria used in this work are listed in Table 1. Cultures stored at −20° were streaked on Chromocult Tryptone Bile X-Glucuronide (TBX) agar plates (Merck, Darmstadt, Germany) and incubated ad 37 °C for 24 h. Subsequently, a single colony was transferred into 10 mL of Luria Bertani (LB) broth (Alfa Aesar, Karlsruhe, Germany), incubated under the same conditions, and used for the following experiments. 

### 4.2. Bacteriophage Propagation

Bacteriophages LF2, FM9, FM10, DP16, DP17 and DP19 [45] were replicated through the double layer method described by Carey-Smith et al. [46] and partially modified. Briefly, 100 μL of bacteriophage from a single plaque were mixed with 1 mL of bacterial culture of CNCTC6896 (Czech National Collection of Type Cultures) strain in exponential phase (OD600nm = 0.2–0.3), used as indicator. After ten minutes, 5 mL of LB soft agar (0.5%) and 40 μL of CaCl_2_ 1 M were gently mixed and poured into LB agar (1.5%) plate. Following an overnight incubation at 37 °C, 4 mL of SM buffer (100 mM NaCI, 8 mM MgSO_4_, 50 mM Tris-HCI, pH 7.5, 0.01% gelatin) were added to each plate. After one hour at room temperature, the buffer and the soft agar layer were collected in a 50 mL sterile tube. The suspension was then centrifuged at 4500 *g* for 10 min (Rotina 380 R, Hettich, Tuttlingen, Germany). The supernatant was transferred in a new tube; the centrifugation was repeated twice and then filtered through a 0.45 μm membrane (Minisart, Sartorius™), transferred in an ultracentrifugation tube (Quick-Seal^®^ Round-Top Polypropylene Tube, Beckman Coulter^®^, Cassina de’ Pecchi, Italy) and centrifuged a 100,000 *g* for 1 h at 4 °C (Beckman Coulter L7-65, Ultracentrifuge). The supernatant was discarded, and the pellet resuspended in 3 mL of SM buffer. After overnight incubation at 4 °C, the dissolved pellet was filtered through a 0.45 μm membrane (Minisart, Sartorius^®^) and stored at 4 °C. 

### 4.3. Biofilm Formation Assay

A crystal violet staining assay was performed to obtain a semi-quantitative determination of biofilm formation in STEC strains [23]. Bacteria were grown overnight in an M9 salts medium (3.39% Na_2_HPO_4_, 1.5% KH_2_PO_4_, 0.25% NaCl and 0.5% NH_4_Cl) supplemented with 0.5% glucose and in TSB (Tryptic Soy Broth) (Scharlab, Sentmenat, Spain) medium at 30 °C and 37 °C without shaking, in 96 wells polystyrene microtiter plates (Starlab, Hamburg, Germany). After incubation, a first reading (OD_600nm_) was performed through a plate reader (PowerWave XS2, BioTek, Winooski, VT, USA) using Gen5. To assess the number of attached cells, the supernatant of each well was discarded, washed twice with sterile water, and then stained with 1% crystal violet for 20 min. Afterward, wells were washed again with sterile water and allowed to dry. To calculate the number of stained cells, the wells were filled with 200 μL of 95% ethanol by vigorous pipetting and the Optical Density (OD_600nm_) was measured again. The experiment was conducted in triplicate. Values of biofilm formation (BF) were calculated according to the following equation as proposed by Naves et al. [22]: BF = AB/CW where AB is the OD_600nm_ of stained attached bacteria and CW is the OD_600nm_ value of stained control wells containing only bacteria-free medium. BF values were classified into four categories according based on the amount of biofilm produced: strong (S): ≥ 6, moderate (M): 5.99 ≥ BF ≥ 4, Weak (W): 3.99 ≥ BF ≥ 2 and negative (N): <2. 

### 4.4. Test for Estimating Biofilm Prevention

Six bacteriophages alone and two mixtures of bacteriophages, named cocktails, were tested for their capacity to prevent biofilm formation. The bacteriophages were: LF2, FM9, FM10, DP16, DP17 and DP19. Cocktail 1 consisted of FM10, DP16 and DP19. Cocktail 2 was composed by all the six phages. The viral titer of each bacteriophage in the cocktails was the same. The behavior of four pathogenic *E. coli* strains ED56, C679-12, ED226 and ED33 belonging to serogroups O26, O104, O113 and O139, respectively, was investigated; two *E. coli* strains CNCTC6246 and ACTC25404 were used as the control. The effect of three different Multiplicities of Infection (MOI) 1, 2, 10 and 100, were analyzed. 

Bacterial culture (100 μL) was inoculated in LB broth in a flat-bottomed 96 well-plate (Porvair Sciences Limited, UK); when, at the exponential phase (OD_600nm_ ≅ 0.2), 100 μL of bacteriophages suspended in SM buffer were added at different concentrations, according to the different MOI. No phage was added to the control. After 24 h of incubation at 30°C, a first reading was performed to ensure that each bacterium had grown regularly. Then, the planktonic cells were removed and washed twice with distilled water. Cells attached to the surface were stained with crystal violet (1%) for 20 min. Afterwards, wells were washed twice with distilled water and the stained cells were dissolved in 200 μL of 95% ethanol. Then, the cell concentration was measured at OD_600nm_ (PowerWave XS2, BioTek, Winooski, VT, USA) using the software Gen5 and compared with the phage-free control. The experiment was performed in triplicate.

### 4.5. Removal of the Formed Biofilm

For the removal of formed biofilm, a phage cocktail containing LF2, FM9, FM10, DP16, DP17 and DP19 was used. The different target biofilms consisted of four pathogenic *E. coli*: ED56, C679-12, ED226, ED33 belonging to O26, O104, O113 and O139 serogroups, respectively, and one non-pathogenic and high biofilm producer *E. coli* strain (ATCC25404). Briefly, a pre-inoculum was prepared in LB broth; subsequently, 50 μL of bacterial culture was dispersed on membranes with a diameter of 25 mm and 0.45 μm pore size (Whatman™ Nuclepore Track-Etched Membranes) which were then placed on TSA plates. After incubation at 30 °C for 24 h, the membranes were transferred to a new plate. Then, 25 μL of phage cocktail at concentration of 1 × 10^8^ PFU/mL, divided in 5 drops of 5 μL, were spotted on the surface of the membrane and incubated under the same conditions. As a control, 25 μL of SM buffer were spotted on membranes without adding phage solution. Then, the membranes were transferred to a tube containing 5 mL of Phosphate-Buffered Saline (PBS) (137 mM NaCl, 2.7 mM KCl, 8 mM Na_2_HPO_4_, and 2 mM KH_2_PO_4_) and vigorously vortexed to remove the formed biofilm. Successively, the membranes were discarded, and the biofilm was homogenized for 90 s (IKA T 10 basic ULTRA-TURRAX^®^). Samples were then diluted in PBS and aliquots of appropriate dilutions were plated on TSA plates. The plates were incubated for 16 h at 37 °C. The test was conducted in three independent experiments. Colonies were counted and expressed as CFU/cm^2^.

The analysis of variance was performed using the open-source software: R Core Team [47], packages: “agricolae”.

### 4.6. Relative Quantification of Stx

For Stx quantification, a commercial ELISA kit (RIDASCREEN Verotoxin kit R-biopharm, Darmstadt, Germany) was used according to the manufacturer’s instructions with slight modifications. Strains ED56 and ED226, possessing genes for Shiga toxins (stx1 and stx2, respectively, data not shown), were grown overnight at 37 °C in Luria Bertani (LB) broth. An aliquot of 100 μL of the overnight cultures was inoculated in tubes containing 10 mL of LB broth and after reaching the exponential phase (OD_600nm_ = 0.2), aliquots of 100 µL were transferred in a fresh tube containing 10 mL of LB broth added with phage Cocktail 2 at MOI 1, prepared as described in 4.4, or with 100 μL of 50 μg/mL mitomycin C (Sigma-Aldrich, St. Louis, MO, USA). A control consisting of pure strain (non-exposed culture) was also prepared. After an overnight incubation at 37 °C, the samples were centrifuged at 6000 rcf for 10 min at room temperature and then filtered through a 0.22 µm membrane filter (Minisart Syringe filter). A standard curve was made with the non-exposed supernatants of each bacterial culture. For each condition, the samples were run in triplicate in the same ELISA plate. To calculate the fold change relative to the non-exposed sample, the mean of the blanks was subtracted from the mean of the OD for each sample, and afterward each mean was multiplied by the respective dilution factor and divided by that of its respective non-exposed sample.

## Figures and Tables

**Figure 1 antibiotics-10-01423-f001:**
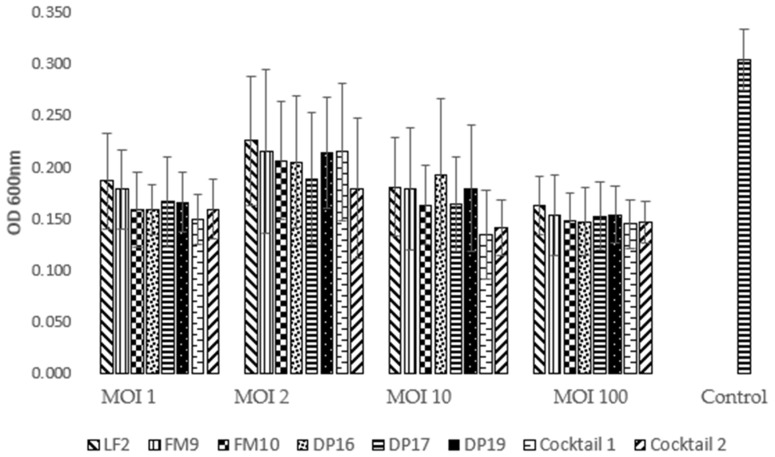
Effect of bacteriophages on biofilm production at different MOI values (average value of OD_600nm_ on the six different *E. coli* strains).

**Figure 2 antibiotics-10-01423-f002:**
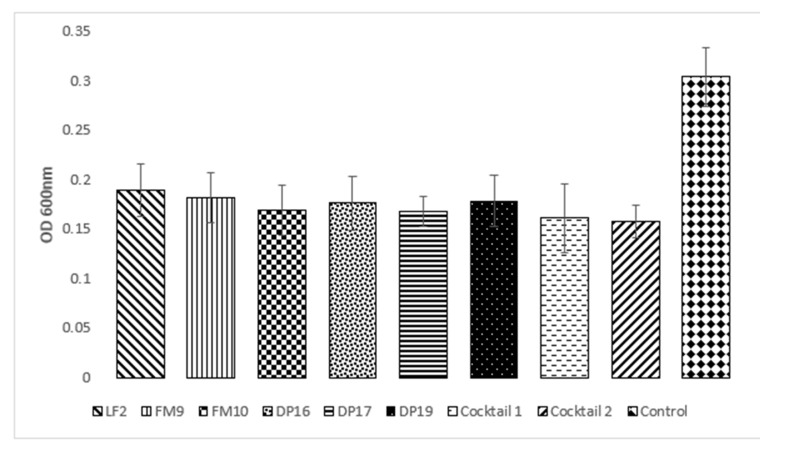
Average of different MOI values compared to the control.

**Figure 3 antibiotics-10-01423-f003:**
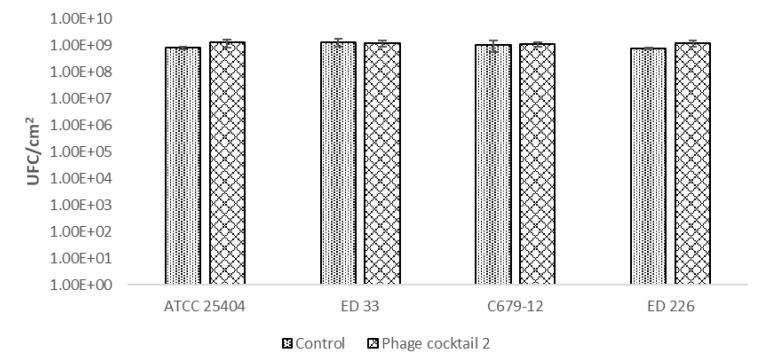
Effect of the phage cocktail on already formed *E. coli* biofilm.

**Figure 4 antibiotics-10-01423-f004:**
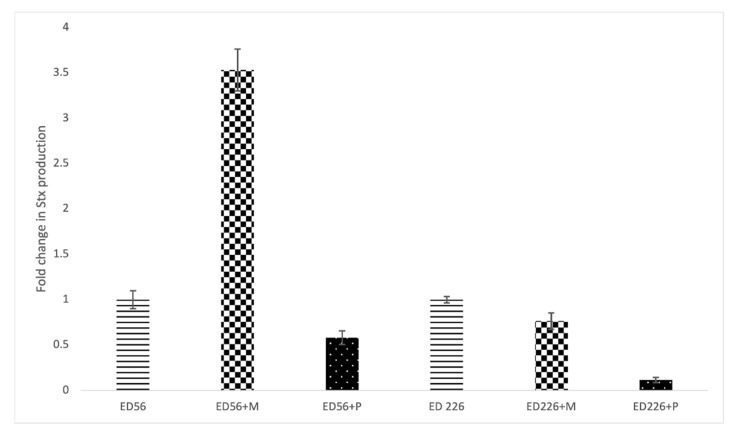
Stx production in cultures exposed to Mitomycin C (+M) or to phage Cocktail2 (+P) expressed as fold change relative to non-exposed cultures (ED56 and ED226).

**Table 1 antibiotics-10-01423-t001:** Biofilm values of investigated STEC strains. Navy blue: strong biofilm producers; azure: moderate biofilm producers; light blue: weak biofilm producers; no color: negative.

Strain	Serogroup	TSB 30 °C	TSB 37 °C	M9 30 °C	M9 37 °C
ATCC35150 ^a^	O157	1.59	1.12	2.09	1.10
393 ^b^	O26	2.37	2.25	1.26	1.06
15R ^b^	O76	1.97	1.10	1.30	1.26
33C ^b^	O23	3.91	3.95	1.32	1.44
380USA ^b^	O157	2.58	3.65	1.36	0.96
62 19/L ^b^	O157	2.19	2.38	3.02	1.23
214CH ^c^	O157	1.62	1.26	1.16	0.94
228GS ^c^	O145	2.90	1.38	1.22	1.08
229RACH ^c^	O111	1.12	1.22	1.32	1.25
239RA ^c^	O26	1.48	1.06	1.27	1.06
243RACH ^c^	O26	1.68	1.01	1.36	1.08
243ROI-A ^c^	O26	2.60	1.70	1.56	1.14
F1-1 ^c^	O26	1.37	0.93	1.43	1.09
F10-4 ^c^	O26	1.96	1.29	1.51	1.63
F11-4 ^c^	O26	2.17	1.20	1.64	1.07
F95 ^c^	O26	6.77	4.83	9.87	7.15
F95-3 ^c^	O26	1.44	1.37	1.41	1.10
PO128 ^c^	O128	1.36	1.02	1.25	1.32
6182-50 ^d^	O113	2.40	2.31	2.50	1.62
C679-12 ^e^	O104	1.80	6.06	1.29	0.87
ED13 ^f^	O157	1.14	0.92	1.49	0.81
ED142 ^f^	O111	1,71	2.45	1.70	1.16
ED161 ^f^	O86	1.98	2.14	1.29	1.20
ED172 ^f^	O103	3.83	1.87	1.62	1.29
ED173 ^f^	O145	2.90	1.98	1.33	1.33
ED226 ^f^	O113	4.30	3.60	1.44	1.68
ED33 ^f^	O139	2.62	1.54	1.95	1.66
ED56 ^f^	O26	2.37	2.28	1.28	1.16
ED76 ^f^	O91	4.51	3.29	1.21	1.22
ED82 ^f^	O111	1.27	0.77	1.29	0.95
ED238 ^f^	O121	1.96	3.44	1.24	1.13

^a^ American Type Culture Collection ^b^ Istituto di Ispezione degli Alimenti di Origine Animale (Milan, Italy); ^c^ [23] ^d^ Collaborative Centre for Reference and Research on Escherichia (WHO) [24] ^e^ Statens Serum Institut (SSI) in Denmark ^f^ Istituto Superiore di Sanità (Rome, Italy).

**Table 2 antibiotics-10-01423-t002:** Value of different MOI in biofilm prevention tests. Values with different letters are significantly different groups (*p* < 0.05) assigned by One-Way Anova (Analysis of Variance) with post hoc Tukey HSD (Honestly Significant Difference).

Phage/Cocktail	MOI 1	MOI 2	MOI 10	MOI 100	Average
LF2	0.187	0.226	0.181	0.163	0.189 ± 0.03
FM9	0.179	0.215	0.179	0.154	0.182 ± 0.03
FM10	0.159	0.206	0.164	0.148	0.169 ± 0.03
DP16	0.159	0.205	0.193	0.147	0.176 ± 0.03
DP17	0.167	0.189	0.164	0.153	0.168 ± 0.02
DP19	0.166	0.214	0.179	0.154	0.178 ± 0.03
Cocktail 1	0.150	0.215	0.135	0.145	0.161 ± 0.04
Cocktail 2	0.159	0.180	0.142	0.147	0.157 ± 0.02
Average	0.166 ^cd^	0.206 ^b^	0.167 ^c^	0.151^d^	* 0.304 ^a^
	±0.012	±0.015	±0.02	±0.06	±0.12

* This value represents the average of six bacterial strains used without any bacteriophage addition.

## Data Availability

The data presented in this study are available in Appendix A.

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
