# Peer review of "Application of Bacteriophages on Shiga Toxin-Producing *Escherichia coli* (STEC) Biofilm"

_antibiotics, 2021, doi:10.3390/antibiotics10111423_

Round 1

Reviewer 1 Report

The manuscript entitled “
Application of bacteriophages on Shiga toxin-producing Escherichia coli (STEC) biofilm.” is discussing an interesting technique for antagonism of biofilms which are considered among the most important virulence factors that make the organism more resistant.
The manuscript is concise, easy to be understandable for readers. The manuscript is also well written: experimental results are adequately described, their interpretation is reasonable and organized discussion. However, I have some key comments, to mention:
For example:
Introduction:
Lines 39-42: Is literature 5 releavant to the wole sentence?
Line 82: Please write Biofilm (BF).
Line 110: Do the authors think that a cocktail of 6 bacteriophages is still useful? There was no big difference in results and this point was not mentioned in the discussion. This leads to a save for time and cost and ignoring the cocktail at all.
Line 114: What is MOI? please mention
Line 141: Is reference 25 relevant in the results? do the authors intend to confirm the theory in this previous publication?
Discussion:
Line 172-175: Other causes were mentioned previously to elucidate this phenomenon, please mention.
Line 200: Are the authors able to test this modification after using bacteriophage depolymerases?

I would recommend minor revisions before publishing the manuscript.

Author Response

Dear Reviewer we took into consideration all of your comments and we answered to your questions:

Lines 39-4: Is literature 5 releavant to the wole sentence? We followed the suggestion and delete reference 5
Line 82: Pease write Biofilm (BF). Done
Line 110:Do the authors think that a cocktail of 6 bacteriophages is still useful? There was no big difference in results and this point was not mentioned in the discussion. This leads to a save for time and cost and ignoring the cocktail at all. We add a comment in discussion about the use of cocktails .
Line 114: What is MOI? please mention Done on line 102
Line 141: Is reference 25 relevant in the results? do the authors intend to confirm the theory in this previous publication? We removed the reference
Discussion:
Line 172-175: Other causes were mentioned previously to elucidate this phenomenon, please mention. Done
Line 200: Are the authors able to test this modification after using bacteriophage depolymerases? We would like to deepen the subject together with the use of depolymerases and other methods to improve biofilm removal but not for this paper

Reviewer 2 Report

This work have a great issue.
It is relevant to know that bacteriophage it is not good enough to destroy biofilm. However, authors are studying to use bacteriophage to destroy biofilm of STEC. STEC strains have a bacteriophage that is the responsible for Stx expression. 
Authors have to perform an assay in which they demonstrate that bacteriophages are not inducing Stx. 
They are talking about the importance of STEC in foods and could be a huge health problem if the treatment of STEC biofilm induce Stx expression.

As conclusion, it is necessary to perform an assay analyzing Stx expression after treatment with bacteriophages.

Author Response

Authors have to perform an assay in which they demonstrate that bacteriophages are not inducing Stx. 
They are talking about the importance of STEC in foods and could be a huge health problem if the treatment of STEC biofilm induce Stx expression.

As conclusion, it is necessary to perform an assay analyzing Stx expression after treatment with bacteriophages. We made the assay and add protocols, results and discussion about that

Round 2

Reviewer 2 Report

Authors responded correctly the comments previously done